# Effect of Hyacinth Treatment by Hydrogen Peroxide Stabilized with Silver and Some Fungicides on the Fungal Infection of Substrate and Bulbs and on Plant Growth and Development

Adam T. Wojdyła [1,*], Jacek S. Nowak [1] , Jan Bocianowski [2] , Jacek Wiśniewski [3] and Emilia Waszkiewicz [4]

1 The National Institute of Horticultural Research, Konstytucji 3 Maja1/3 Str., 96-100 Skierniewice, Poland
2 Department of Mathematical and Statistical Methods, Poznań University of Life Sciences, Wojska Polskiego 28, 60-637 Poznań, Poland
3 Gospodarstwo Ogrodnicze Jacek Wiśniewski Spółka Jawna, Góraszka 87, 05-462 Wiązowna, Poland
4 Emi Agro Emilia Waszkiewicz, Kościuszki 11/2 Str., 05-400 Otwock, Poland
* Correspondence: adam.wojdyla@inhort.pl; Tel.: +48-46-834-67-61

**Abstract:** The aim of the study was to demonstrate the feasibility of using hydrogen peroxide stabilized with silver ($H_2O_2$-$Ag^+$) and selected fungicides for the treatment of hyacinth bulbs and to determine their effects on the development of fungi colonizing the substrate and bulbs and their impact on plant growth and development. Hydrogen peroxide stabilized with silver ($H_2O_2$-$Ag^+$) and captan, pyraclostrobin + boscalid and methyl thiophanate + tetraconazole were used to treat hyacinth bulbs before planting, in the form of a 20 min soak. The effect of the treatment on the development of fungi on the bulbs and substrate was evaluated during rooting in the refrigerated storerooms and after placement in the greenhouse. Observations were also made on the effect of treatment on hyacinth growth and development. The study showed a significant effect of $H_2O_2$-$Ag^+$ and fungicides used for the treatment of hyacinth on reducing the development of fungi on bulbs and substrate during the period of hyacinth rooting. It was found that $H_2O_2$-$Ag^+$, at concentrations ranging from 2% to 10%, significantly increased the Chlorophyll Index and Nitrogen Balance Index. In some of the concentrations tested, it also increased petal width, inflorescence width, the number of flowers, leaf length, leaf width, plant quality, fresh weight without inflorescences and the dry weight of plants without inflorescences at some of the concentrations tested. It was shown that, for all fungicides used, flower diameter, inflorescence width, total height, leaf length, leaf width, plant fresh weight without inflorescence and plant dry weight without inflorescence were significantly increased compared to control plants. None of the treatments tested were phytotoxic to hyacinth.

**Keywords:** bulb sprouting; hydrogen peroxide stabilized with silver; fungicides; inhibition; fungi on bulb and substrate surface; plant growth and development

## 1. Introduction

The literature data indicate that the most common causes of hyacinth bulb rot are the fungi *Penicillium* spp., *Aspergillus* spp., *Rhizopus stolonifer* (Ehrenb.) Vuill. and *Sclerotinia* spp. [1] and the bulb base rot of *Fusarium oxysporum* Schldl [1,2]. Occasionally, the causer of bulb rot is the bacterium *Erwinia carotovora* (*Pectobacterium carotovorum* Hauben) [1]. On the farm presented here, in the case of the annual forcing of several million hyacinth bulbs, even such a small percentage of diseased bulbs carries significant material losses. There are no data in the available literature on the possibility of using protection products to treat hyacinth bulbs. Observations to date have shown that, for plants propagated by seed, seed treatment is the most effective and cost-efficient method of protection and brings the greatest reduction in resource use and financial outlay [3], especially as the European Commission plans to reduce pesticide use by 50% by 2030. This implies the need to search for other methods of protection to replace the chemicals withdrawn from use.

In addition to preventing or reducing bulb rot, it is very important to determine the effect of the treatment agents on the development of fungi on the bulb surface and substrate during the rooting of bulbs. High humidity above 80% during rooting causes the fungi *Neopestalotiopsis foedans* (Sacc. & Ellis) Maharachch. to develop on the bulb surface and substrate, along with K.D. Hyde & Crous (syn. *Pestalotiopsis foedans* (Sacc. & Ellis) Steyaert)), *Penicillium olsonii* Bainier & Sartory, *Rhizopus stolonifer* (Ehrenb.) Vuill. and *Trichoderma asperellum* Samuels, Lieckf. & Nirenberg, sometimes covering more than 80% of their surface with abundant mycelium and spores [4].

*1.1. Influence of Treatment of Hyacinth Bulbs by Hydrogen Peroxide Stabilized with Silver and Fungicides on the Coverage of Bulbs and Substrate with Mycelium*

The mechanism of the bactericidal action of hydrogen peroxide is based on its breakdown inside the cell by protective enzymes (e.g., catalase) and/or metal ions-reducers present there ($Fe^{2+}$, $Cu^+$). As a result of the decomposition reaction of $H_2O_2$, unstable and highly reactive hydroxyl radicals (HO) are formed, reacting with nucleophiles present in the cell. As a result, DNA strands are broken, resulting in mutations of the genetic material. Hydroxyl radicals can also damage cell membranes [5]. The results of model studies on this issue have shown a complete degradation of the lipid membrane after 17 min of exposure to radicals [6]. In the case of anaerobic organisms, the main cytotoxic agent is the end product of hydrogen peroxide decomposition—molecular oxygen [7]. $H_2O_2$ molecules, due to their small size, easily penetrate cell membranes, showing biocidal activity against a wide range of organisms—bacteria, fungi and viruses [8]. Hydrogen peroxide ($H_2O_2$) is defined as reactive oxygen species (ROS) capable of damaging various cellular structures. On the other hand, recent work has shown that $H_2O_2$ can also act as a potent signaling molecule, mediating various physiological and biochemical processes in plants [9].

Fallik et al. [10] showed that $H_2O_2$-$Ag^+$, at a concentration of 0.5%, applied to a pepper fruit bath before storage, reduced fruit rot, with an almost complete inhibition of pathogens such as *Botrytis cinerea* Pers. and *Alternaria alternata* (Fr.) Keissl. Soaking potato tubers (Spunta variety) in solutions of hydrogen peroxide (0, 20, 40, 60 and 80 mM) before planting did not improve tuber yield [11]. However, the use of hydrogen peroxide at low concentrations (20 mM) reduces the chlorophyll content of potato leaves [11].

Hydrogen peroxide in in vitro tests reduced the mycelial growth, spore formation and spore germination of *Rhizopus oryzae* Went & H.C. Prinsen Geerligs, the causer of tobacco leaf rot [12], and *Penicillium expansum* Link, the causer of apple rot in storage [13]. EL-Mougy et al. [14] showed in vitro that hydrogen peroxide at concentrations of 1.5 and 2% completely inhibited the linear growth and spore germination of *B. cinerea* Pers., *R. stolonifer* (Ehrenb.) Vuill., *P. digitatum* Sacc. and *P. italicum* Wehmer isolates obtained from strawberry and orange. Additionally, hydrogen peroxide in vitro showed a high efficacy in limiting the growth of *Pestalotia psidii* Pat., a fungus isolated from guava fruit [15].

Silver has strong antimicrobial activity in both ionic and nanoparticle forms; thus, it has applications in water sanitation, the control of plant diseases caused by biotic and abiotic agents and for the sterilization of medical instruments [16]. Lamsal et al. [17] showed that, under in vitro conditions, silver nanoparticles at a concentration of 100 ppm maximally inhibited the mycelial growth and spore germination of various *Colletotrichum* species. Under field conditions, nanoparticle silver applied prophylactically at a concentration of 50 ppm proved to be highly effective against anthracnose on pepper. On the other hand, Abdelmalek and Salaheldin [18] found under in vitro conditions that silver nanoparticles (150 ppm) had a very strong antifungal effect on *Alternaria alternata* (Fr.) Keissl., *A. citri* Ellis & N.Pierce and *Penicillium digitatum* Sacc.

Previous studies have shown that captan and boscalid + pyraclostrobin (Signum 33 WG), applied during the growing season, were highly effective in preventing grapevine fruit rot caused by *Penicillium expansum* Link and *Rhizopus stolonifer* (Ehrenb.) Vuill. A particularly high efficacy was found with their prophylactic application [19]. In contrast, the occurrence of grey mold (*Botrytis cinerea* Pers.) and wet rot of the fruit (*Rhizopus stolonifer*

(Ehrenb.) Vuill.) is a serious problem during the storage period of strawberries, reducing the quality and quantity of the yield. Boscalid + pyraclostrobin (Signum 33 WG), used to protect strawberries between flowering and fruit harvest, significantly reduced the occurrence of the disease on stored fruit [20]. Other authors also confirm the high efficacy of the fungicide Signum 33 WG applied to strawberry sprays in reducing the spread of *Rhizopus* and *Mucor* fungi to other fruit [21].

Captan, on the other hand, in in vitro laboratory studies, proved highly effective in reducing the mycelial growth and spore germination of *R. oryzae* Went & H.C. Prinsen Geerligs [22]. Similarly, under in vivo conditions, captan showed a high efficacy in reducing rot caused by *R. oryzae* Went & H.C. Prinsen Geerligs. Prinsen Geerligs on potato [22] and aubergine [15] (Youssef et al. 2015). Bhale and Rajkonda [23], in in vitro studies, showed that captan in vitro at concentrations above 500 μg/mL reduced the growth of *Trichoderma* spp. isolates (*T. viride* Pers., *T. harzianum* Rifai, *T. koningii* Oudem., *T. pseudokoningii* Rifai and *T. virens* Mill., Giddens & A.A. Foster Arx). On the other hand, Ray et al. [24], in *in vitro* studies on PDA (potato dextrose agar), found that Topsin M (thiophanate methyl) at a concentration of 0.1% to 0.2% caused the complete inhibition of the mycelial growth of *Pestalotiopsis disseminata* (Thüm.) Steyaert.

*1.2. Influence of the Treatment of Hyacinth Bulbs by Hydrogen Peroxide Stabilized with Silver and Fungicides on Plant Growth and Quality*

There is a lack of information in the available literature on the effects of the tested products on the growth and development of hyacinth. The available data on the effect of agents on plant growth and development are often contradictory and dependent on the plant species to which it was applied. Therefore, another very important issue undertaken in our study was to determine the effect of the applied measures on the growth and development of hyacinth. The literature data indicate that soaking bread wheat seeds in different concentrations of $H_2O_2$ solution increased the density of stomata and the length and histological components of the leaf. These changes in histological components appeared to be positive, as plants from seeds soaked in $H_2O_2$ (pre-treatment with $H_2O_2$) had a higher fresh and dry weight, a higher number of seeds per spike and a higher grain yield [25]. Lopez-Delgado et al. [26] showed that spraying potatoes with hydrogen peroxide, starting from the 3rd to the 10th week of their cultivation, resulted in an approximately 30% increase in shoot thickening, a significant increase in the lignin content of the conductive bundles and an increase in their dimensions. In addition, the authors found a 3–4-fold increase in starch and about a 60% increase in lignin content in the stems of plants sprayed with the test agent. In contrast, Jung et al. [27] found in greenhouse studies that Ag nanocolloids showed a positive effect on the growth of the fresh and dry weight of onion plants.

In the case of fungicides, a previous study by Tort and Turkyilmaz [28] showed that captan applied to the seed treatment of *Capsicum annuum* L. pepper reduced the seed germination rate. On the other hand, the authors showed that, when used for seedling spraying, it caused an increase in chlorophyll *a* and *b* content in the leaves. Similarly, Seyhan et al. [29] in in vitro laboratory tests showed negative effects of captan on seed germination and pepper growth. Studies by Manjunath and Bagyaraj [30] do not confirm the negative effect of captan applied to the soil at low doses of 2.5 μg/g of soil on onion growth. On the other hand, in the field cultivation of soybean, pyraclostrobin can be successfully applied to spray plants to increase root biomass and improve shoot and leaf growth, nitrogen assimilation and, consequently, the yield itself [31].

The aim of the study was to demonstrate the feasibility of using hydrogen peroxide stabilized with silver ($H_2O_2$-$Ag^+$) and selected fungicides for the treatment of hyacinth bulbs and to determine their effects on the development of fungi colonizing the substrate and bulbs and their impact on plant growth and development.

## 2. Materials and Methods

### 2.1. Growing Media Preparation and Their Physical and Chemical Properties

The substrate for hyacinth rooting was prepared at the Gospodarstwo Ogrodnicze Jacek Wiśniewski Spółka Jawna, Góraszka. The substrate consisted of deacidified high peat and sand mixed in a 9:1 ratio. Physical and chemical analyses of the materials used for composting were performed at the Chemical Analysis Laboratory, Institute of Horticulture in Skierniewice (Poland) and are summarized in Table 1. The pH and the electrolytic conductivity (EC) of the substrates were determined in a suspension of the substrate and distilled water (*v:v*; 1:3). The moisture content, bulk density, air capacity, water capacity, shrinkage and total porosity were assessed according to UE methods [32]. The content of absorbable macronutrients was determined with the universal method [32,33] used in Poland for the analysis of horticultural soils and growing media. The extraction solution included 0.03 N acetic acid (pH 3.2), and the medium ratio was 1:10 (*v:v*).

**Table 1.** Physical and chemical properties of the substrate used in the experiment.

| Parameters | Value |
|:---:|:---:|
| Total pore space (%) | 92.1 |
| Bulk density (g·cm$^{-3}$) | 0.15 |
| Shrinkage (%) | 21.2 |
| Water volume at −10 cm $H_2O$ (%) | 75.0 |
| Air volume at −10 cm $H_2O$ (%) | 17.1 |
| Organic matter (%) | 59.5 |
| Ash content (%) | 40.5 |
| pH ($H_2O$, 1:3) | 6.5 |
| EC (mS·cm$^{-1}$) | 0.14 |
| N-NO$_3$ (mg·dm$^{-3}$) | 6.0 |
| P (mg·dm$^{-3}$) | 23.0 |
| K (mg·dm$^{-3}$) | 14.0 |
| Mg (mg·dm$^{-3}$) | 259.0 |
| Ca (mg·dm$^{-3}$) | 719.0 |
| Na (mg·dm$^{-3}$) | 32.5 |
| S (mg·dm$^{-3}$) | 16.0 |
| Cl (mg·dm$^{-3}$) | 15.2 |
| Fe (mg·dm$^{-3}$) | 28.1 |
| Mn (mg·dm$^{-3}$) | 3.74 |
| Cu (mg·dm$^{-3}$) | 0.32 |
| Zn (mg·dm$^{-3}$) | 3.61 |
| B (mg·dm$^{-3}$) | 0.36 |

### 2.2. Experimental Design

The experiment was carried out in 2021–2022 at the Gospodarstwo Ogrodnicze Jacek Wiśniewski Spółka Jawna and the Institute of Horticulture—National Research Institute in Skierniewice. The trials on the treatment of hyacinth bulbs included the plant growth stimulator Bisteran (50% hydrogen peroxide + 0.32 g silver in 1 kg) and the fungicides Biszop 80 WG (80% captan), Signum 33 WG (67 g pyraclostrobin in 1 kg + 267 g boscalid in 1 kg) and Yamato 303 SE (233 g thiophanate-methyl in 1 L + 70 g tetraconazole in 1 L).

*Hyacinthus orientalis* 'Blue Pearl' bulbs in prepared 3 kg raschel bags were immersed in the test media for 20 min according to the following scheme for each combination:

1. Control
2. Bisterane (hydrogen peroxide stabilized with silver—$H_2O_2$-Ag$^+$) 1%.
3. Bisterane (hydrogen peroxide stabilized with silver—$H_2O_2$-Ag$^+$) 2%.
4. Bisterane (hydrogen peroxide stabilized with silver—$H_2O_2$-Ag$^+$) 3%.
5. Bisterane (hydrogen peroxide stabilized with silver—$H_2O_2$-Ag$^+$) 5%.
6. Bisterane (hydrogen peroxide stabilized with silver—$H_2O_2$-Ag$^+$) 10%.
7. Yamato 303 SE 0.5%.

8. Signum 33 WG 0.5%.
9. Biszop 80 WG 1%.

To soak the bulbs, a 10 l of preparation suspension was prepared in plastic buckets. The bags with the bulbs were then removed, and, after draining off the excess liquid and drying the decomposed bulbs for several minutes, the bulbs were planted into pots with a diameter of 8 cm and a volume of 0.3 L. The plants were planted into the medium described in Section 2.1. Hyacinths with a bulb circumference of 17–18 cm were planted into the substrate so that 2/3 of the bulb was visible above the surface of the substrate. The pots with the bulbs were then placed in plastic boxes measuring $60 \times 40 \times 20$ cm (length $\times$ width $\times$ height). The boxes in which the pots with planted hyacinths were placed were covered with a $60 \times 40 \times 2.5$ cm upholstery sponge and then pressed with $60 \times 40$ cm multiplates to prevent the bulbs from being pushed out of the substrate by the developing root system. The plastic boxes together with the planted bulbs were placed on pallets with five boxes in 10–11 layers per level. The wooden pallets together with the bulbs were placed in refrigerated storerooms for a hyacinth period of 12 weeks. After insertion into the refrigerated room, the temperature was maintained at 9 °C for a period of 12 weeks. In turn, the humidity in the storerooms was maintained at 94–99% throughout the period. The humidity in the storerooms was maintained using a special computer program developed for this purpose. According to the computer, a fogging system was activated usually every 10–14 days for a period of two hours, with high-powered fans installed to distribute the fog thoroughly throughout the storeroom. On one occasion, 10 liters of water were used to aerosolize one 600 $m^3$ storeroom. In one refrigerated storeroom of this capacity, four pallets were placed, with 45 plastic crates with pots on each pallet. After four weeks of the experiment, the pressure sponge cover was removed from the crates containing the hyacinth bulbs.

*2.3. Measurements and Observations*

During the course of the experiment, observations were made after 4, 8 and 12 weeks and after the completion of bulb rooting (bulbs covered with mycelium and substrate mycelium coverage). Then, after the aboveground parts had been cut, observations were made of the coverage of the bulbs and the substrate by the fungi most commonly isolated in earlier observations: *Neopestalotiopsis foedans* (syn. *Pestalotia foedans*, *Pestalotiopsis foedans*), *Penicillium olsonii*, *Rhizopus stolonifer* and *Trichoderma asperellum* [4].

After the rotting bulbs were removed from the cold storage (second stage), they were placed in the greenhouse of the Institute of Horticulture on climate-controlled flooding tables, where a temperature of around 9 °C was maintained for several days (5–7 days) for acclimatization. After this time, the greenhouse temperature was raised to 16 °C and maintained until the end of cultivation. When necessary, the plants were watered so that the surface of the substrate in the pot was slightly moist. During cultivation in the greenhouse, observations were made of the occurrence of microorganisms on the surface of the substrate and the hyacinth bulbs, and the onset of flowering of the plants was recorded. Prior to biometric measurements, chlorophyll, flavonoid and nitrogen balance index (NBI) contents were determined using a Dualex 4 m. At full flowering, biometric measurements were taken, i.e., total height, green mass height, the number of leaves per plant, leaf length, leaf width, the number of flowers per plant, the number of buds per plant, flower diameter, flower height, the length of individual flower petals, the width of individual flower petals, the length of corolla (trumpet), the width of corolla (trumpet), the fresh weight of the plant without inflorescence, the dry weight of the plant without inflorescence, the fresh weight of flowers and the dry weight of flowers. A qualitative assessment of the plants was also made on a scale of 1–5 pts, where: 1 pt—microorganisms on the surface of the substrate and hyacinth bulbs, poorly developed inflorescence; 3 pts—traces of microorganisms on the surface of the substrate and on the hyacinth bulbs, well-developed inflorescence; 5 pts—no microorganisms on the substrate surface and hyacinth bulbs, very good habit and fully developed inflorescence. The root system of the plants was also assessed on a scale of

1–5 pts, where 1 pt—no visible roots; 3 pts—good rooting, but roots do not form a compact mass; 5 pts—very good rooting, very well-formed root ball.

### 2.4. Statistical Analysis

### 2.4.1. Statistical Calculations for Mycelial Coverage of Bulbs and Substrate

The normality of the distribution of the observed traits was verified with Shapiro–Wilk's normality test. One-way (combinations) analyses of variance (ANOVA) were performed to verify the null hypotheses of a lack of a combination effect on the two observed traits (bulbs covered with mycelium and substrate mycelium coverage) in three terms (after 4 weeks, after 8 weeks and after 12 weeks), independently for each one. The arithmetic means and standard deviations were calculated. Moreover, Fisher's least significant differences (LSDs) were estimated at a significance level of $\alpha = 0.05$. The relationships between observed traits were estimated using Pearson's linear correlation coefficients. The GenStat v. 18 statistical software package (VSN International) was used for the analyses.

### 2.4.2. Statistical Calculations for Plant Growth and Quality Traits

The normality of the distribution of the 19 traits, i.e., petal length—PL, petal width—PW, flower diameter—FD, inflorescence height—IH, inflorescence width—IW, flower number—FN, fresh mass of flowers—FMF, dry mass of flowers—DMF, total height—TH, leaf number—LN, leaf length—LL, leaf width—LW, quality evaluation of the plant—QEP, fresh mass of the aboveground part (without flowers)—FMAP, dry weight of the aboveground part (without flowers)—DWAP, starting height—SH, chlorophyll Index DUALEX—CLD, flavonoid Index DUALEX—FLD and Nitrogen Balance Index—NBI, was verified with Shapiro–Wilk's normality test. The homogeneity of variance was tested using Bartlett's test. Box's M test was used to check the multivariate normality and homogeneity of variance–covariance matrices. All the traits had a normal distribution. A one-way (combinations) multivariate analysis of variance (MANOVA) was performed. Following this, one-way analyses of variance (ANOVA) were performed to verify the null hypotheses of a lack of a combination effect on the 19 observed traits, independently for each one. The arithmetic means and standard deviations were calculated. Moreover, Fisher's least significant differences (LSDs) were estimated at a significance level of $\alpha = 0.05$. The relationships between observed traits were estimated using Pearson's linear correlation coefficients. The results were also analyzed using multivariate methods: (1) principal component analysis (PCA) was applied to present a multi-trait assessment of the similarity of the tested combinations in a lower number of dimensions with the least possible loss of information, and (2) the differences among the analyzed combinations were verified by cluster analysis using the nearest neighbor method and Euclidean distances [34]. The GenStat v. 18 statistical software package (VSN International) was used for the analyses.

## 3. Results

### 3.1. Coverage of Bulbs and Substrate with Mycelium after the Treatment of Hyacinth Bulbs by Hydrogen Peroxide Stabilized with Silver and Fungicides

All of the observed traits had a normal distribution. ANOVA indicated that the main effect of combinations was significant for all examined traits (Table 2). The mean values, standard deviations (s.d.), Fisher's least significant differences (LSD) and homogeneous groups of observed traits for combinations are presented in Table 2. The highest mean values of observed traits (bulbs mycelium coverage and substrate mycelium coverage, in all three terms) were observed for combination No. 1—control. On the other hand, the lowest mean values of bulbs mycelium coverage were observed for combination No. 5 (after 4 weeks) and combination No. 6 (after 8 and 12 weeks). In the case of substrate mycelial coverage, combinations No. 4 and 8 were characterized by the lowest mean values after 4 weeks (mean values equal to 0.00). Combination No. 4 had the smallest mean value of substrate mycelial coverage also after 8 weeks (0.0625) and 12 weeks (0.00). After 12 weeks, mean values equal to zero were also obtained for combinations No. 2 and 5 (Table 2).

The studies also revealed a significant correlation between observed traits. The significant positive correlations were observed between all pairs of traits, except for: PPG 4 w—0.461, PPG 4 w—CPG 8 w, PPG 4 w—CPG 12 w and PPG 4 w—PPG 12 w (Table 3, Figure 1).

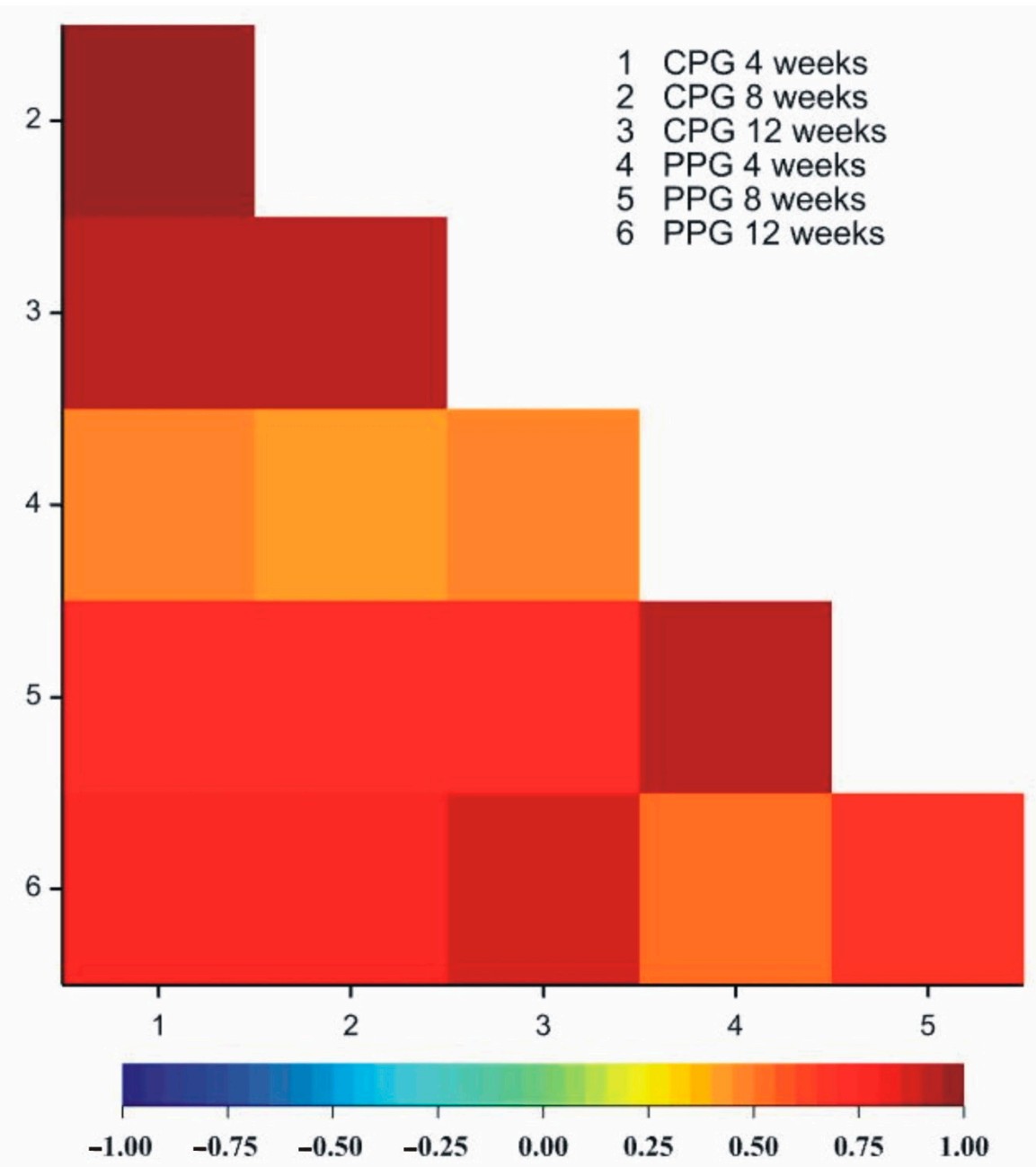

**Figure 1.** Heatmaps for linear Pearson's correlation coefficients between the observed traits: CPG and PPG in 4, 8 and 12 weeks.

**Table 2.** Results of one-way analysis of variance, mean values and standard deviations (s.d.), as well as homogeneous groups for observed traits.

| Combinat. | Bulbs Covered with Mycelium | | | | | | Substrate Mycelium Coverage | | | | | |
|---|---|---|---|---|---|---|---|---|---|---|---|---|
| | After 4 Weeks | | After 8 Weeks | | After 12 Weeks | | After 4 Weeks | | After 8 Weeks | | After 12 Weeks | |
| | Mean | s.d. | Mean | s.d. | Mean | s.d. | Mean | s.d. | Mean | s.d. | Mean | s.d. |
| 1 | 29.75 a | 18.47 | 26 a | 11.997 | 5.813a | 6.082 | 1.4375 a | 4.149 | 3 a | 5.372 | 0.875 a | 2.843 |
| 2 | 18.04 b | 14.46 | 11.312 b | 7.784 | 1.5 cde | 3.593 | 1.125 ab | 3.18 | 1.9375 b | 3.937 | 0 b | 0 |
| 3 | 8.5 de | 9.39 | 4.562 c | 5.857 | 0.75 def | 2.651 | 0.125 cd | 1.118 | 0.25 c | 1.355 | 0.125 b | 1.118 |
| 4 | 5.69 ef | 9.77 | 5.125 c | 6.161 | 1.5 cde | 3.68 | 0 d | 0 | 0.0625 c | 0.559 | 0 b | 0 |
| 5 | 2.87 f | 5.14 | 3.812 c | 5.75 | 0.438 ef | 1.629 | 0.125 cd | 1.118 | 0.3125 c | 1.658 | 0 b | 0 |
| 6 | 3.37 f | 6.74 | 0.875 d | 3.058 | 0.25 f | 1.097 | 0.5 bcd | 2.333 | 0.375 c | 1.546 | 0.125 b | 1.118 |
| 7 | 11.28 cd | 7.59 | 8.875 b | 7.247 | 1.75 cd | 4.141 | 0.75 bc | 2.767 | 0.6875 c | 2.214 | 0.3125 b | 1.658 |
| 8 | 29.13 a | 16.03 | 25.062 a | 16.506 | 4 b | 5.921 | 0 d | 0 | 0.6875 c | 2.066 | 0.375 b | 1.738 |
| 9 | 12.13 c | 8.34 | 3.75 c | 4.673 | 2.062 c | 4.262 | 0.125 cd | 1.118 | 0.1875 c | 1.244 | 0.25 b | 1.097 |
| LSD$_{0.05}$ | 3.57 | | 2.67 | | 1.245 | | 0.688 | | 0.817 | | 0.434 | |
| *F*-ANOVA | 63.32 *** | | 94.56 *** | | 16.28 *** | | 4.57 *** | | 11.24 *** | | 3.20 ** | |

** $p < 0.01$; *** $p < 0.001$; a, b, c,...—in columns, means followed by the same letters are not significantly different.

**Table 3.** Correlation coefficients between observed traits.

| Trait [1] | CPG 4 w | CPG 8 w | CPG 12 w | PPG 4 w | PPG 8 w | PPG 12 w |
|---|---|---|---|---|---|---|
| CPG 4 w | 1 | | | | | |
| CPG 8 w | 0.961 *** | 1 | | | | |
| CPG 12 w | 0.918 *** | 0.926 *** | 1 | | | |
| PPG 4 w | 0.461 | 0.432 | 0.455 | 1 | | |
| PPG 8 w | 0.722 * | 0.711 * | 0.711 * | 0.909 *** | 1 | |
| PPG 12 w | 0.757 * | 0.768 * | 0.899 *** | 0.542 | 0.682 * | 1 |

* $p < 0.05$; *** $p < 0.001$. [1] CPG 4 w—bulbs covered with mycelium after 4 weeks; CPG 8—bulbs covered with mycelium after 8 weeks; CPG 12—bulbs covered with mycelium after 12 weeks; PPG 4 w—substrate coverage with mycelium after 4 weeks; PPG 8 w—substrate coverage with mycelium after 8 weeks; PPG 12 w—substrate coverage with mycelium after 12 weeks (PPG 12 w).

### 3.2. Plant Growth and Quality after the Treatment of Hyacinth Bulbs by Hydrogen Peroxide Stabilized with Silver and Fungicides

All the observed traits had a normal distribution. The results of the MANOVA indicated that the effect of combinations (Wilk's λ = 0.0144; $F$ = 7.84; $p < 0.0001$) was significantly different regarding all the 19 quantitative traits. ANOVA indicated that the main effect of combinations was significant for all examined traits, except for PW, DMF, LN and QEP (Table 4).

Mean values, standard deviations (s.d.), Fisher's least significant differences (LSD) and homogeneous groups of observed traits for combinations are presented in Table 5. PL ranged from 11.66 (for combination 3) to 12.32 (for combination 9), PW ranged from 6.912 (for combination 9) to 9.012 (for combination 3), FD ranged from 4.216 (for combination 2) to 4.639 (for combination 9), IH ranged from 3.665 (for combination 4) to 3.818 (for combination 9), IW ranged from 0.6576 (for combination 1) to 0.79 (for combination 8), FN ranged from 19.12 (for combination 3) to 21.95 (for combination 7), FMF ranged from 11.61 (for combination 3) to 13.19 (for combination 7), DMF ranged from 1.137 (for combination 8) to 1.242 (for combination 9), TH ranged from 21.26 (for combination 3) to 23.31 (for combination 9), LN ranged from 5.637 (for combination 2) to 5.825 (for combination 4), LL ranged from 13.23 (for combination 1) to 15.2 (for combination 9), LW ranged from 2.108 (for combination 1) to 2.302 (for combination 7), QEP ranged from 4.35 (for combination 9) to 5.1 (for combination 2), FMAP ranged from 20.48 (for combination 1) to 23.27 (for combination 9), DWAP ranged from 1.55 (for combination 1) to 1.733 (for combination 8), SH ranged from 1.46 (for combination 3) to 1.94 (for combination 1), CLD ranged from 45.69 (for combination 1) to 50.58 (for combination 9), FLD ranged from 1.479 (for combination 9) to 1.608 (for combination 2) and NBI ranged from 29.45 (for combination 1) to 34.39 (for combination 9) (Table 5).

**Table 4.** Mean squares (m.s.) from one-way analysis of variance for observed traits.

| Trait (Short Key) | Source of Variation | Combinations | Residual |
|---|---|---|---|
| Petal length (PL) | d.f. | 8 | 710 |
|  | m.s. | 4.028 * | 1.991 |
| Petal width (PW) | d.f. | 8 | 710 |
|  | m.s. | 28.24 | 15.6 |
| Flower diameter (FD) | d.f. | 8 | 710 |
|  | m.s. | 1.4476 *** | 0.3172 |
| Inflorescence height (IH) | d.f. | 8 | 710 |
|  | m.s. | 0.1942 * | 0.09716 |
| Inflorescence width (IW) | d.f. | 8 | 710 |
|  | m.s. | 0.188537 *** | 0.008382 |
| Flower number (FN) | d.f. | 8 | 351 |
|  | m.s. | 63.74 *** | 17.46 |
| Fresh mass of flowers (FMF) | d.f. | 8 | 351 |
|  | m.s. | 25.249 *** | 4.321 |
| Dry mass of flowers (DMF) | d.f. | 8 | 670 |
|  | m.s. | 0.08985 | 0.0598 |
| Total height (TH) | d.f. | 8 | 668 |
|  | m.s. | 43.054 *** | 6.996 |
| Leaf number (LN) | d.f. | 8 | 710 |
|  | m.s. | 0.5359 | 0.3553 |
| Leaf length (LL) | d.f. | 8 | 710 |
|  | m.s. | 34.383 *** | 4.638 |
| Leaf width (LW) | d.f. | 8 | 710 |
|  | m.s. | 0.32609 *** | 0.06746 |
| Quality evaluation of plant (QEP) | d.f. | 8 | 709 |
|  | m.s. | 3.862 | 2.496 |
| Fresh mass of the above-ground part (without flowers) (FMAP) | d.f. | 8 | 710 |
|  | m.s. | 74.5 *** | 11.92 |
| Dry weight of the above-ground part (without flowers) (DWAP) | d.f. | 8 | 710 |
|  | m.s. | 0.26687 *** | 0.06831 |
| Starting height (SH) | d.f. | 6 | 133 |
|  | m.s. | 2.217 *** | 0.1337 |
| Chlorophyll Index DUALEX (CLD) | d.f. | 8 | 351 |
|  | m.s. | 76.82 *** | 17.63 |
| Flavonoid Index DUALEX (FLD) | d.f. | 8 | 351 |
|  | m.s. | 0.05422 * | 0.02188 |
| Nitrogen Balance Index (NBI) | d.f. | 8 | 351 |
|  | m.s. | 90.76 *** | 13.7 |

* $p < 0.05$; *** $p < 0.001$; d.f.—the number of degrees of freedom.

**Table 5.** Mean values, standard deviations (s.d.), Fisher's least significant differences (LSD) and homogeneous groups of observed traits for combinations.

| Trait | Combination | 1 | 2 | 3 | 4 | 5 | 6 | 7 | 8 | 9 | LSD$_{0.05}$ |
|---|---|---|---|---|---|---|---|---|---|---|---|
| EN | Mean | 12.21 a | 12.21 a | 11.66 c | 11.97 abc | 11.98 abc | 12.08 abc | 11.73 bc | 12.16 ab | 12.32 a | 0.44 |
| | s.d. | 1.131 | 1.466 | 1.234 | 1.387 | 1.519 | 1.813 | 1.214 | 1.389 | 1.429 | |
| PW | Mean | 7.337 b | 7.338 b | 9.012 a | 7.4 b | 7.406 b | 7.494 b | 7.38 b | 7.175 b | 6.912 b | 1.23 |
| | s.d. | 0.61 | 0.482 | 10.323 | 0.536 | 0.552 | 0.542 | 5.649 | 0.516 | 0.605 | |
| FD | Mean | 4.362 cd | 4.216 d | 4.395 bcd | 4.432 bc | 4.462 abc | 4.509 abc | 4.624 a | 4.559 ab | 4.639 a | 0.18 |
| | s.d. | 0.7667 | 0.4799 | 0.456 | 0.6607 | 0.573 | 0.5039 | 0.6033 | 0.511 | 0.4305 | |
| IH | Mean | 3.704 bc | 3.697 bc | 3.697 bc | 3.665 c | 3.682 bc | 3.761 abc | 3.722 abc | 3.769 ab | 3.818 a | 0.1 |
| | s.d. | 0.5638 | 0.2625 | 0.2598 | 0.2921 | 0.2333 | 0.2905 | 0.2535 | 0.2519 | 0.2605 | |
| IW | Mean | 0.6576 f | 0.7363 bc | 0.6988 de | 0.6779 ef | 0.6756 ef | 0.7546 b | 0.7146 cd | 0.79 a | 0.7863 a | 0.03 |
| | s.d. | 0.09829 | 0.1043 | 0.07919 | 0.09318 | 0.07291 | 0.09642 | 0.09227 | 0.09284 | 0.09054 | |
| FN | Mean | 19.62 cd | 19.67 cd | 19.12 d | 20.37 bcd | 19.93 bcd | 21.05 ab | 21.95 a | 19.42 cd | 20.65 abc | 1.3 |
| | s.d. | 5.559 | 4.066 | 3.131 | 4.106 | 3.198 | 4.397 | 5.038 | 4.42 | 2.923 | |
| FMF | Mean | 12.33 cd | 11.82 cde | 11.61 e | 12.42 bc | 11.81 cde | 12.38 bc | 13.19 a | 11.71 de | 12.99 ab | 0.65 |
| | s.d. | 2.421 | 2.247 | 1.995 | 2.017 | 2.2 | 1.947 | 1.984 | 1.887 | 1.95 | |
| DMF | Mean | 1.221 a | 1.218 a | 1.226 a | 1.214 a | 1.174 a | 1.205 a | 1.241 a | 1.137 a | 1.242 a | 0.77 |
| | s.d. | 0.3283 | 0.1982 | 0.2823 | 0.2339 | 0.182 | 0.2426 | 0.2214 | 0.227 | 0.2618 | |
| TH | Mean | 22.09 c | 22.17 bc | 21.26 d | 21.56 cd | 21.83 cd | 21.69 cd | 23.14 a | 22.93 ab | 23.31 a | 0.82 |
| | s.d. | 3.198 | 2.524 | 2.209 | 2.342 | 2.516 | 2.891 | 2.32 | 3.207 | 2.208 | |
| LN | Mean | 5.687 a | 5.637 a | 5.75 a | 5.825 a | 5.638 a | 5.638 a | 5.646 a | 5.8 a | 5.813 a | 0.19 |
| | s.d. | 0.8656 | 0.5335 | 0.5156 | 0.4975 | 0.6005 | 0.6005 | 0.621 | 0.5372 | 0.5055 | |
| LL | Mean | 13.23 e | 13.58 e | 13.81 de | 13.81 de | 14.27 cd | 14.41 bcd | 14.52 bc | 15.01 ab | 15.2 a | 0.67 |
| | s.d. | 2.374 | 1.887 | 1.714 | 1.806 | 3.714 | 1.956 | 1.78 | 1.452 | 1.854 | |
| LW | Mean | 2.108 d | 2.226 abc | 2.153 cd | 2.161 cd | 2.176 bcd | 2.228 abc | 2.302 a | 2.283 a | 2.245 ab | 0.08 |
| | s.d. | 0.308 | 0.1892 | 0.1783 | 0.1777 | 0.1726 | 0.1768 | 0.5394 | 0.1524 | 0.2018 | |
| QEP | Mean | 4.481 b | 5.1 a | 4.569 b | 4.588 b | 4.581 b | 4.462 b | 4.52 b | 4.381 b | 4.35 b | 0.49 |
| | s.d. | 0.5363 | 4.537 | 0.4411 | 0.4555 | 0.4317 | 0.5017 | 0.4996 | 0.4587 | 0.4865 | |
| FMAP | Mean | 20.48 d | 21.01 cd | 21.02 cd | 21.44 cd | 21.45 cd | 21.91 bc | 22.6 ab | 22.98 ab | 23.27 a | 1.08 |
| | s.d. | 3.781 | 3.924 | 2.83 | 3.538 | 2.961 | 3.826 | 3.122 | 3.338 | 3.563 | |
| DWAP | Mean | 1.55 d | 1.625 cd | 1.602 cd | 1.657 abc | 1.613 cd | 1.65 bc | 1.659 abc | 1.733 a | 1.724 ab | 0.08 |
| | s.d. | 0.2806 | 0.2632 | 0.1978 | 0.3662 | 0.187 | 0.2515 | 0.2102 | 0.3177 | 0.2236 | |
| SH | Mean | 1.94 a | 1.77 b | 1.46 d | 1.605 c | 1.785 b | 1.91 a | 1.765 b | | | 0.11 |
| | s.d. | 0.347 | 0.3672 | 0.2836 | 0.3677 | 0.3233 | 0.4789 | 0.3617 | | | |
| CLD | Mean | 45.69 d | 47.06 cd | 48.1 bc | 49.05 ab | 48.66 bc | 48.37 bc | 47.09 cd | 48.24 bc | 50.58 a | 1.85 |
| | s.d. | 4.375 | 4.504 | 4.425 | 3.281 | 3.087 | 3.696 | 5.017 | 4.212 | 4.777 | |
| FLD | Mean | 1.539 ab | 1.608 a | 1.533 b | 1.513 b | 1.501 b | 1.525 b | 1.524 b | 1.495 b | 1.479 b | 0.07 |
| | s.d. | 0.1643 | 0.1539 | 0.1526 | 0.1264 | 0.1385 | 0.1233 | 0.1579 | 0.1448 | 0.1632 | |
| NBI | Mean | 29.87 cd | 29.45 d | 31.64 b | 32.58 b | 32.7 b | 31.88 b | 31.16 bc | 32.39 b | 34.39 a | 1.63 |
| | s.d. | 3.967 | 3.377 | 4.115 | 2.866 | 3.743 | 3.14 | 4.12 | 4.348 | 3.358 | |

a, b, c, . . .—in rows, means followed by the same letters are not significantly different.

The studies also revealed significant correlation between observed traits. The positive correlations were observed between the following pairs of traits: FD-LL (0.86), FD-FMAP (0.87), FD-DWAP (0.67), FD-NBI (0.72), IH-IW (0.86), IH-TH (0.68), IH-LL (0.79), IH-FMAP (0.79), IH-DWAP (0.70), IW-LL (0.79), IW-LW 90.76), IW-FMAP (0.79), IW-DWAP (0.84), FN-FMF (0.87), TH-LL (0.68), TH-LW (0.74), TH-FMAP (0.78), LL-LW (0.76), LL-FMAP (0.97), LL-DWAP (0.89), LL-CLD (0.69), LL-NBI (0.76), LW-FMAP (0.83), LW-DWAP (0.77), FMAP-DWAP (0.93), FMAP-NBI (0.68), DWAP-CLD (0.69), DWAP-NBI (0.67), and CLD-NBI (0.93). Negative correlations were observed between: PL-PK (−0.74), PW-SH (−0.67),

FD-QEP (−0.81), FD-FLD (−0.84), LL-FLD (−0.70), QEP-FLD (−0.89), and FLD-NBI (−0.88) (Table 6, Figure 2).

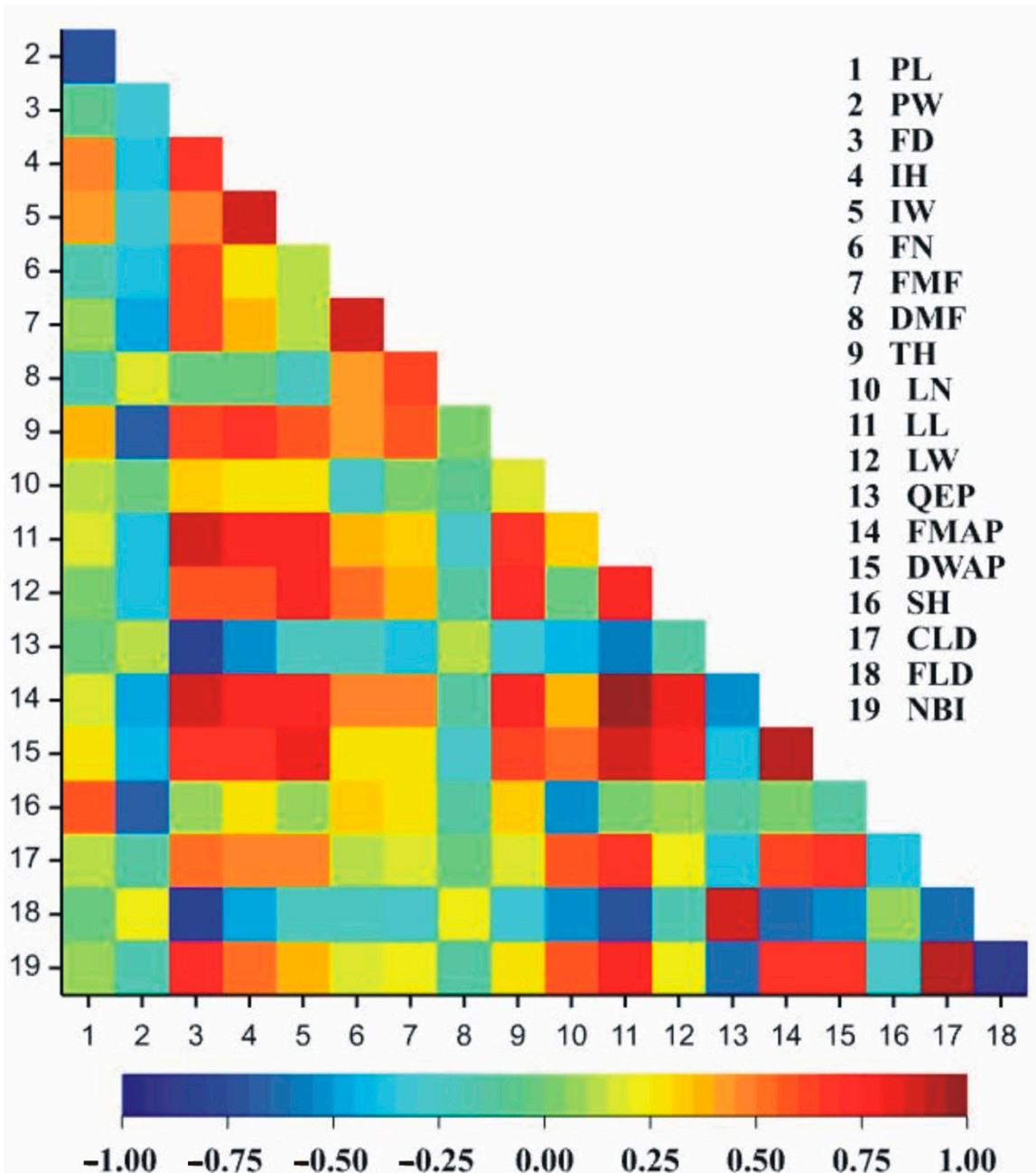

**Figure 2.** Heatmaps for linear Pearson's correlation coefficients between the observed traits: CPG and PPG in 4, 8 and 12 weeks.

Figure 3 shows the variability of nine combinations on the basis of 19 traits in terms of the first two principal components. In the graph, the coordinates of the point for particular combinations are the values for the first and second principal components, respectively. The first two principal components accounted for 84.89% of the total multivariate variability between the individual combinations. Significant positive linear relationships with the first principal component were found for the FD, LL, FMAP, DWAP, CLD and NBI, while the first principal component correlated negatively with FLD (Table 5). The second principal component had a significant positive correlation with FN, FMF and TH (Table 7).

**Table 6.** Correlation coefficients between observed traits.

| Trait | EN | PW | FD | IH | IW | FN | FMF | DMF | TH | LN | LL | LW | QEP | FMAP | DWAP | SH | CLD | FLD |
|---|---|---|---|---|---|---|---|---|---|---|---|---|---|---|---|---|---|---|
| PW | −0.74 * | | | | | | | | | | | | | | | | | |
| FD | −0.08 | −0.31 | | | | | | | | | | | | | | | | |
| IH | 0.49 | −0.4 | 0.65 | | | | | | | | | | | | | | | |
| IW | 0.41 | −0.34 | 0.46 | 0.86 ** | | | | | | | | | | | | | | |
| FN | −0.16 | −0.39 | 0.6 | 0.26 | 0.14 | | | | | | | | | | | | | |
| FMF | 0.05 | −0.47 | 0.61 | 0.38 | 0.11 | 0.87 ** | | | | | | | | | | | | |
| DMF | −0.18 | 0.16 | −0.02 | −0.01 | −0.21 | 0.43 | 0.63 | | | | | | | | | | | |
| TH | 0.37 | −0.66 | 0.62 | 0.68 * | 0.6 | 0.42 | 0.57 | 0.04 | | | | | | | | | | |
| LN | 0.15 | −0.03 | 0.31 | 0.27 | 0.27 | −0.25 | 0.04 | −0.1 | 0.16 | | | | | | | | | |
| LL | 0.16 | −0.38 | 0.86 ** | 0.79 * | 0.79 * | 0.38 | 0.33 | −0.3 | 0.68 * | 0.33 | | | | | | | | |
| LW | 0.03 | −0.39 | 0.59 | 0.58 | 0.76 * | 0.54 | 0.36 | −0.1 | 0.74 * | 0 | 0.76 * | | | | | | | |
| QEP | 0 | 0.12 | −0.81 ** | −0.55 | −0.21 | −0.23 | −0.4 | 0.14 | −0.3 | −0.4 | −0.57 | −0.1 | | | | | | |
| FMAP | 0.17 | −0.46 | 0.87 ** | 0.79 * | 0.79 * | 0.47 | 0.47 | −0.1 | 0.78 * | 0.38 | 0.97 *** | 0.83 ** | −0.54 | | | | | |
| DWAP | 0.26 | −0.44 | 0.67 * | 0.70 * | 0.84 ** | 0.29 | 0.27 | −0.3 | 0.64 | 0.54 | 0.89 ** | 0.77 * | −0.37 | 0.93 *** | | | | |
| SH | 0.6 | −0.67 * | 0.06 | 0.28 | 0.08 | 0.31 | 0.28 | −0.1 | 0.3 | −0.5 | 0.01 | 0.1 | −0.1 | 0.02 | −0.12 | | | |
| CLD | 0.14 | −0.15 | 0.52 | 0.46 | 0.48 | 0.14 | 0.16 | 0 | 0.15 | 0.57 | 0.69 * | 0.24 | −0.35 | 0.61 | 0.69 * | −0.4 | | |
| FLD | −0.01 | 0.21 | −0.84 ** | −0.46 | −0.23 | −0.21 | −0.3 | 0.25 | −0.32 | −0.5 | −0.70 * | −0.16 | 0.89 ** | −0.64 | −0.53 | 0.07 | −0.64 | |
| NBI | 0.08 | −0.19 | 0.72 * | 0.5 | 0.4 | 0.19 | 0.25 | −0.1 | 0.26 | 0.6 | 0.76 * | 0.23 | −0.65 | 0.68 * | 0.67 * | −0.3 | 0.93 *** | −0.88 ** |

* $p < 0.05$; ** $p < 0.01$; *** $p < 0.001$.

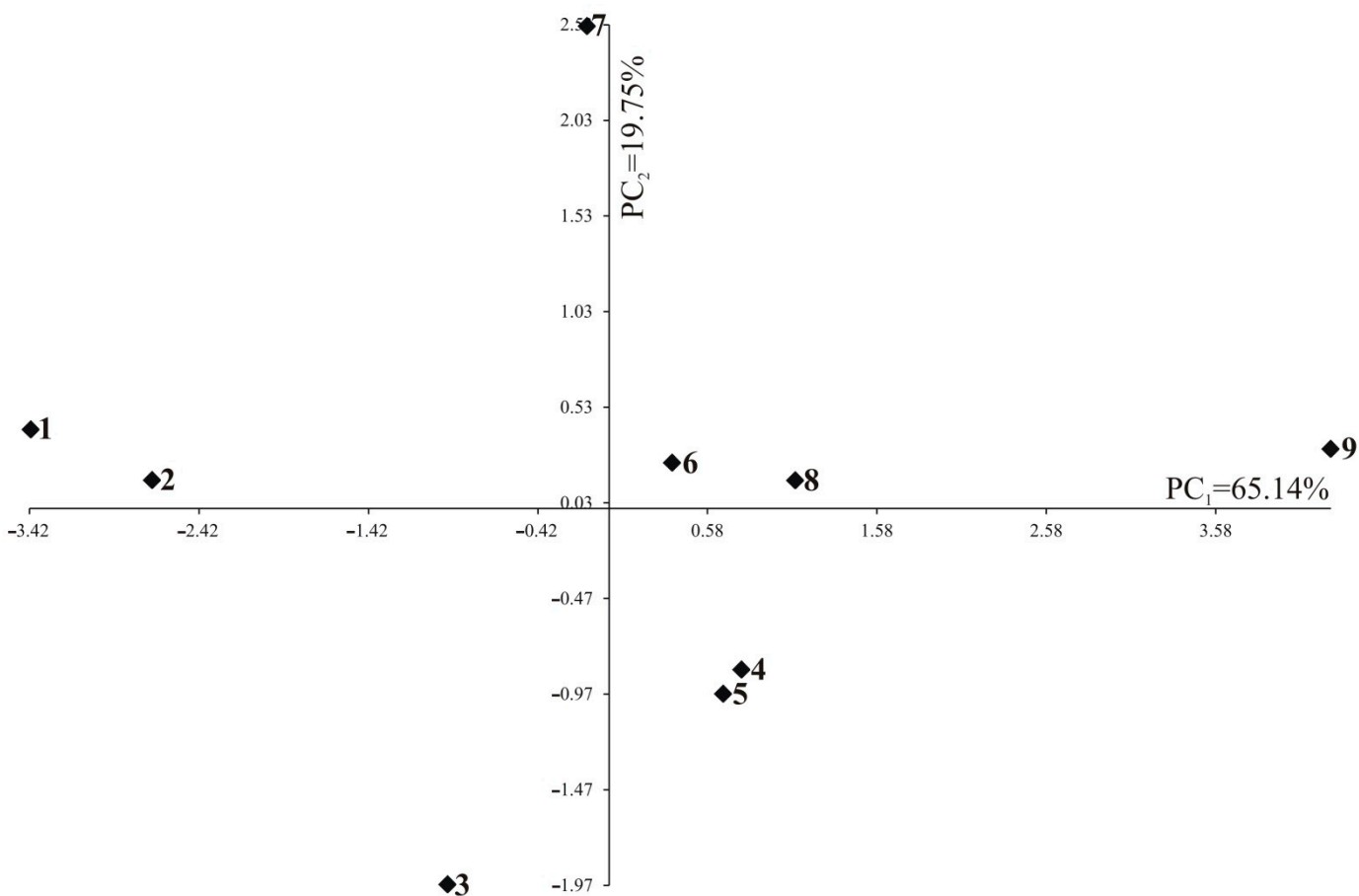

**Figure 3.** Distribution of combinations in space of the first two canonical variables. 1—control; 2—Bisterane (hydrogen peroxide stabilized with silver—$H_2O_2$-$Ag^+$) 1%; 3—Bisterane (Silver stabilized hydrogen peroxide—$H_2O_2$-$Ag^+$) 2%; 4—Bisterane (hydrogen peroxide stabilized with silver—$H_2O_2$-$Ag^+$) 3%; 5—Bisterane (hydrogen peroxide stabilized with silver—$H_2O_2$-$Ag^+$) 5%; 6—Bisterane (hydrogen peroxide stabilized with silver—$H_2O_2$-$Ag^+$) 10%; 7—Yamato 303 SE 0.5%; 8—Signum 33 WG 0.5%; 9—Biszop 80 WG 1%.

**Table 7.** Correlation coefficients between the first two canonical variates and studied traits.

| Trait | PC$_1$ | PC$_2$ |
|---|---|---|
| EN | 0.151 | 0.132 |
| PW | −0.322 | −0.582 |
| FD | 0.786 * | 0.427 |
| IH | 0.629 | 0.351 |
| IW | 0.572 | 0.255 |
| FN | 0.342 | 0.737 * |
| FMF | 0.375 | 0.725 * |
| DMF | −0.075 | 0.246 |
| TH | 0.439 | 0.740 * |
| LN | 0.542 | −0.32 |
| LL | 0.873 ** | 0.281 |
| LW | 0.454 | 0.638 |
| QEP | −0.577 | −0.106 |
| FMAP | 0.829 ** | 0.439 |
| DWAP | 0.810 ** | 0.24 |
| SH | −0.18 | 0.567 |
| CLD | 0.927 *** | −0.305 |
| FLD | −0.805 ** | 0.04 |
| NBI | 0.963 *** | −0.205 |
| Percentage of explained multivariate variability | 65.14 | 19.75 |

* $p < 0.05$; ** $p < 0.01$; *** $p < 0.001$.

In the presented dendrogram, as a result of agglomeration grouping using the Euclidean method, all the examined combinations were divided into three groups (Figure 4). The first group included two combinations (No. 8 and No. 9). The second one included only one combination—No. 7. The third one included all the other combinations and was divided into two subgroups: A—no. 1 and no. 2 and B—all other combinations—numbers 3, 4, 5 and 6 (Figure 4).

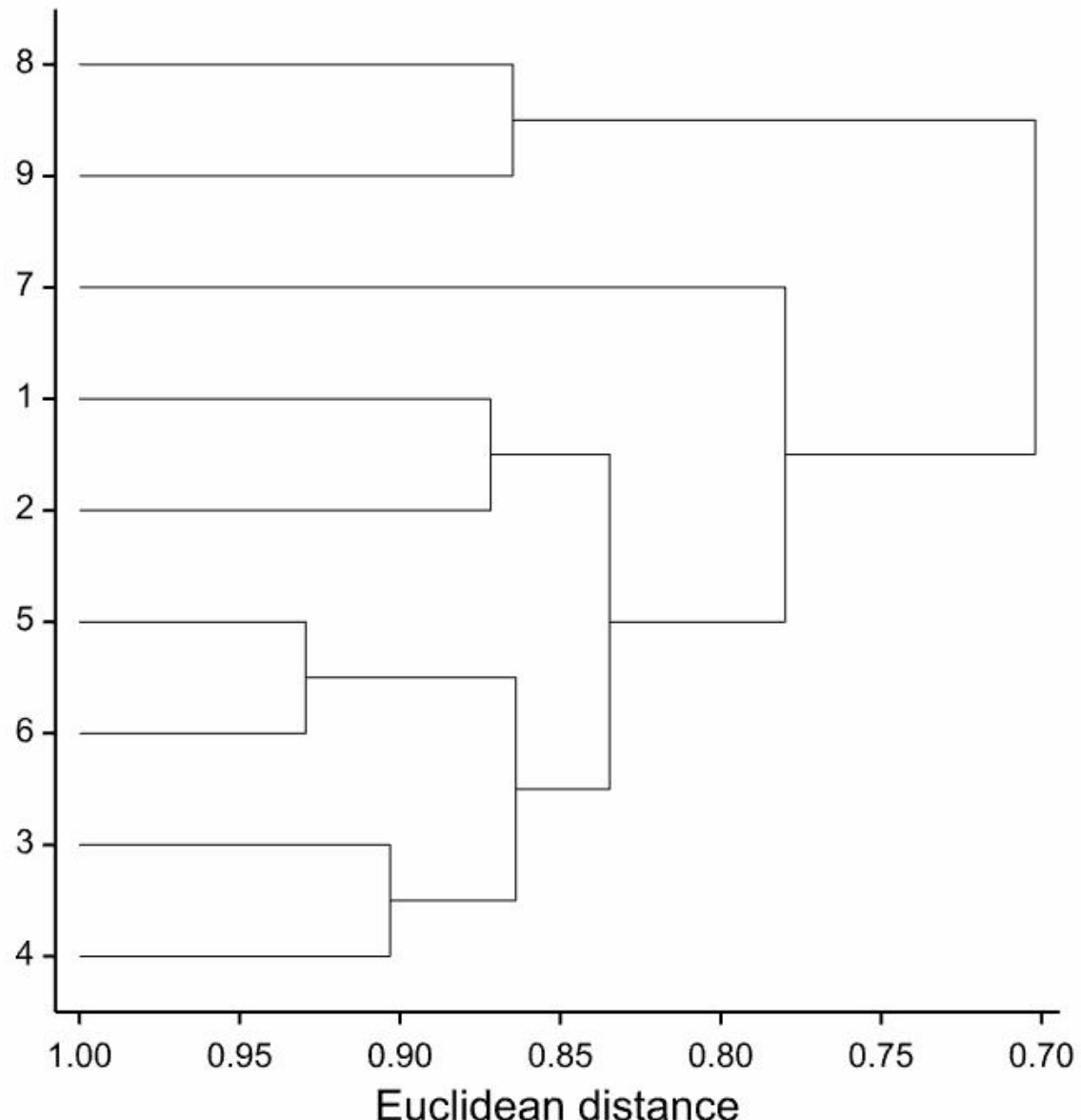

**Figure 4.** Dendrogram of cluster groupings of nine combinations based on all 19 quantitative traits. 1—control; 2—Bisterane (hydrogen peroxide stabilized with silver—$H_2O_2$-$Ag^+$) 1%; 3—Bisterane (hydrogen peroxide stabilized with silver—$H_2O_2$-$Ag^+$) 2%; 4—Bisterane (hydrogen peroxide stabilized with silver—$H_2O_2$-$Ag^+$) 3%; 5—Bisterane (hydrogen peroxide stabilized with silver—$H_2O_2$-$Ag^+$) 5%; 6—Bisterane (hydrogen peroxide stabilized with silver—$H_2O_2$-$Ag^+$) 10%; 7—Yamato 303 SE 0.5%; 8—Signum 33 WG 0.5%; 9—Biszop 80 WG 1%.

## 4. Discussion

*4.1. Effect of the Treatment of Hyacinth Bulbs by Hydrogen Peroxide Stabilized with Silver and Fungicides on the Mycelial Coverage of Bulbs and Substrate*

Given the lack of information in the available literature regarding the possibility of using fungicides to treat hyacinth bulbs, there is a need to carry out such research. The

use of other non-protective and completely environmentally safe compounds, such as hydrogen peroxide, for treating hyacinth bulbs against pathogens seems particularly interesting. Hydrogen peroxide is a popular oxidant and an effective biocide (destroys viruses, bacteria and fungi). It is one of the most 'eco-friendly' chemicals, as the only products of its decomposition are oxygen and water [5]. It is noteworthy that silver-stabilized hydrogen peroxide shows a different mechanism of action on pathogens compared to previously used fungicides and is completely safe for humans and the environment. In recent years, interest in hydrogen peroxide as a disinfectant has increased significantly following reports that toxic organic halogens (including trichloromethane) can be formed during the water chlorination process. The French Ministry of Health has recently approved the use of hydrogen peroxide for drinking water disinfection, and many countries including Switzerland, Germany and Austria have approved the use of a composition composed of silver(I) ions and hydrogen peroxide for this purpose [5].

Unfortunately, in the available literature, there is a lack of research results on the possibility of using treatment products on hyacinth bulbs and their effect on reducing the development of fungi on the bulb and substrate surface and on plant growth and development. Previous in-house research carried out during the hyacinth, narcissus and tulip rooting period showed the fungi *Neopestalotiopsis foedans* (Sacc. & Ellis) Maharachch., K.D. Hyde & Crous, *Penicillium olsonii* Bainier & Sartory, *Rhizopus stolonifer* (Ehrenb.) Vuill. and *Trichoderma asperellum* Samuels, Lieckf. & Nirenberg reducing the quality of commercial material [4]. The conducted studies showed a significant effect of hydrogen peroxide stabilized with silver ($H_2O_2$-$Ag^+$) and fungicides, used for the treatment of hyacinth, on the reduction of fungal growth on bulbs and substrate. The study showed that the percentage mycelial coverage of hyacinth bulbs after 4, 8 and 12 weeks of keeping the plants in cold storage was highest for the control plants. Similar to the control, low efficacy in reducing the fungal coverage of bulbs was found when they were treated in the fungicide pyraclostrobin + boscalid. Bulbs soaked in $H_2O_2$-$Ag^+$ at all tested concentrations and observation dates were significantly in a lower percentage covered by mycelium compared to the control. There was a significant increase in the efficacy of $H_2O_2$-$Ag^+$ in inhibiting fungal growth on bulbs as the concentration used for the treatment increased. Thiophanate-methyl + tetraconazole and captan showed a lower efficacy than $H_2O_2$-$Ag^+$ at concentrations of 2% and higher. In contrast, the mycelial coverage of the potting media in all observations was highest for the control. The lowest effectiveness in reducing mycelial growth on the substrate was found with $H_2O_2$-$Ag^+$ at a concentration of 1%. Hydrogen peroxide stabilized with silver at the other concentrations tested, as well as fungicides, very strongly reduced mycelial development and spore development on the substrate surface.

Our own results obtained from tests on hyacinth are not confirmed in the literature. However, the agents we tested were evaluated by other authors on other plant species and, often, other fungal species. The high efficacy of hydrogen peroxide against *R. stolonifer* (Ehrenb.) Vuill. is confirmed on strawberry [14], boscalid + trifloxystrobin on grapevine [19] and strawberry [20,21]. The high efficacy of captan is confirmed against aubergine fruit rot *R. oryzae* Went & H.C. Prinsen Geerligs [15]. On the other hand, in vitro, hydrogen peroxide [12], captan [35] and trifloxystrobin and thiofanate methyl [36] reduced the mycelial growth of *R. oryzae* Went & H.C. Prinsen Geerligs. Similarly, in our own in vitro studies, captan, boscalid + trifloxystrobin, and trifloxystrobin and tifanate methyl reduced the mycelial growth of *R. stolonifer* (Ehrenb.) Vuill. [4].

In the case of the genus Penicillium under field conditions, a high efficacy of hydrogen peroxide against P. digitatum Sacc. and P. italicum Wehmer on strawberry was demonstrated by [14], along with citrus [37] and pyraclostrobin + boscalid on grapevine against P. expan-sum Link [19]. On the other hand, under in vitro conditions, a direct effect of hydrogen peroxide on limiting the mycelial growth of *P. expansum* Link from apples was confirmed by [13,14,38,39] and Meng et al. [37] in relation to *P. digitatum* Sacc. and *P. italicum* Wehmer. Previous in vitro studies also showed that hydrogen peroxide stabilized with silver at concentrations of 0.05–0.2% strongly inhibited the growth of the mycelial

*Penicillium olsonii* Bainier & Sartory [4]. In contrast, the captan, pyraclostrobin + boscalid and Yamato 303 SE, used in the study in more than 80%, and Yamato 303 SE, used in 43%, reduced the mycelial growth of *P. olsonii* Bainier & Sartory.

Similarly, other authors have demonstrated the high efficacy of the agents used in our experiments in limiting the mycelial growth of various species of the genus *Pestalotia* (*Neopestalotiopsis*). Previous in vitro studies confirm the good efficacy of hydrogen peroxide in limiting the growth of *Pestalotia psidii* Pat., a fungus isolated from guava fruit [15]. Additionally, Ray et al. [24], in an earlier in vitro study, found that Topsin M (thiophanate methyl) at a concentration of 0.1 to 0.2% caused the complete inhibition of the mycelial growth of *Pestalotiopsis disseminata* Causing Grey Blight Disease in Som (*Persea bombycina* Kost.). Laboratory studies showed that $H_2O_2$-$Ag^+$, at concentrations of 0.05–0.2%, strongly inhibited the growth of *Neopestalotiopsis foedans* mycelium [4]. In contrast, captan, pyraclostrobin + boscalid and trifloxystrobin and tifanate methyl, used in the study, reduced mycelial growth by more than 75%.

In the case of the fungus *Trichoderma asperellum* Samuels, Lieckf. & Nirenberg, in their own study, $H_2O_2$-$Ag^+$ at concentrations of 0.05–0.2% and pyraclostrobin + boscalid slightly reduced mycelial growth [4]. In contrast, captan and thiophanate-methyl + tetraconazole reduced mycelial growth by 75%.

Given the use of freshly prepared substrate and containers for planting, it is reasonable to assume that the main source of fungi appearing during the rooting period was the dead arches covering the bulbs. This hypothesis is supported by our own observations. When the entire tulip bulbs were covered with substrate at the time of planting, the problem of fungal incidence during the rooting period was a marginal issue.

### 4.2. Effects of Treating Hyacinth Bulbs by Hydrogen Peroxide Stabilized with Silver and Fungicides on Plant Growth and Quality

Within the 19 traits tested, after soaking the bulbs in the different agents, no differences were found in petal width (PW), flower dry weight (DMF), the number of leaves per plant (LN), plant quality (QEP) and flavonoid index (FLD) compared to control plants. In contrast, significant differences between the combinations were observed for the other traits tested. It was found that $H_2O_2$-$Ag^+$, at concentrations ranging from 2 to 10%, significantly increased the Chlorophyll Index (CLD) and Nitrogen Balance Index (NBI). It also increased the petal width (PW), inflorescence width (IW), number of flowers (FN), leaf length (LL), leaf width (LW), plant quality (QEP), fresh weight without inflorescence (FMAP) and dry weight of plants without inflorescence (DWAP) at some of the concentrations tested.

Flower diameter (FD), inflorescence width (IW), total height (TH), leaf length (LL), leaf width (LW), plant fresh weight without inflorescence (FMAP) and plant dry weight without inflorescence (DWAP) were shown to be significantly higher for all fungicides used compared to control plants. Similarly, boscalid + trifloxystrobin and captan significantly increased the chlorophyll Index (CLD) and Nitrogen Balance Index (NBI). Apart from $H_2O_2$-$Ag^+$ at a concentration of 2% and trifloxystrobine and tifanate methyl limiting flower petal length (PL), all traits tested showed an increase or no difference from the control. This suggests the possibility of using the tested agents to soak the bulbs before planting without fear of phytotoxicity to the hyacinth.

The stimulation of plant growth by some of the agents we tested is confirmed in the literature. Hydrogen peroxide has been shown to be an agent that can stimulate plant growth and development and affect yield increases in potato [11,26], maize [40] and rice [41]. Pyraclostrobin in plants alters metabolism, resulting in increased biomass and yield [42]. Usually, the increase in the yield of plants with disease symptoms after fungicide treatment is interpreted as a pathogen-reducing factor. However, studies conducted on healthy plants sprayed with strobilurin fungicides do not confirm this. In studies, healthy plants on which no pathogen was found sprayed with strobilurins (pyraclostrobin) were more intensely green and looked healthier than plants not treated with a fungicide [43]. In the field, healthy

wheat and barley sprayed with strobilurin (pyraclostrobin) [43] and soybean [44] showed improved growth and yield.

In addition to improving nitrogen metabolism, strobilurins also have an effect on the hormonal balance of plants, as they reduce ethylene synthesis and increase the synthesis and reduce the degradation of cytokinins [45]. As a result, the plants show a reduction in chlorophyll degradation, causing the so-called 'green effect' and delaying leaf yellowing caused by chlorophyll degradation [46,47].

The literature data indicating a negative effect of the agents used on hyacinth growth and development were not confirmed. Tort and Turkyilmaz [28] showed that captan used for the seed treatment of *Capsicum annuum* L. peppers had the effect of reducing the seed germination rate. On the other hand, the authors showed that, when used for seedling spraying, it caused an increase in chlorophyll *a* and *b* content in the leaves. Similarly, Seyhan et al. [29], in in vitro laboratory tests, showed a negative effect of captan on seed germination and pepper growth.

None of the tested agents used for bulb treatment were phytotoxic. Only at the highest concentrations of 5% and 10% ($H_2O_2$-$Ag^+$) was the loss of the intense purple color of the bulbs observed after soaking, while, at 2% to 3%, the loss of the purple color was only partial.

## 5. Conclusions

The aim of the study was to demonstrate the feasibility of using hydrogen peroxide stabilized with silver ($H_2O_2$-$Ag^+$) and selected fungicides for the treatment of hyacinth bulbs and to determine their effects on the development of fungi colonizing the substrate and bulbs and their impact on plant growth and development. The study showed a significant effect of hydrogen peroxide stabilized with silver ($H_2O_2$-$Ag^+$) and fungicides used to treat hyacinth on the reduction of fungal growth on bulbs and substrate during the rooting period. Hydrogen peroxide stabilized with silver and fungicides significantly stimulated the growth and development of the hyacinth in comparison to the control for most of the tested traits. It is important to emphasize that none of the tested products used to treat hyacinth bulbs were phytotoxic. The obtained results may be useful for the practical use of hydrogen peroxide stabilized with silver as a completely safe agent for humans, animals and the environment in the form of the treatment of hyacinth bulbs before rooting in a cold store.

**Author Contributions:** Conceptualization, A.T.W.; methodology, A.T.W., J.S.N., J.B. and J.W.; software, J.B. and J.S.N.; validation, A.T.W., J.S.N., J.W., E.W. and J.B.; formal analysis, J.S.N. and J.B.; investigation, A.T.W., J.S.N., J.W. and E.W.; resources, J.W. and E.W.; data curation, A.T.W.; writing—original draft preparation, A.T.W., J.S.N. and J.B.; writing—review and editing, A.T.W., J.S.N., J.W. and J.B.; visualization, A.T.W., J.S.N. and J.B.; supervision, J.S.N., J.W. and E.W.; project administration, A.T.W., J.W. and E.W.; funding acquisition, A.T.W. and J.W. All authors have read and agreed to the published version of the manuscript.

**Funding:** The research was carried out as part of a research project co-financed by the European Union under the 'Cooperation' measure of the Rural Development Program 2014–2020: 'Implementation of an improved product, innovative technology and production organization methods in the production of bulbous ornamental plants using high-pressure fogging of refrigerated rooms with silver-stabilized hydrogen peroxide'.

**Institutional Review Board Statement:** Not applicable.

**Informed Consent Statement:** Not applicable.

**Data Availability Statement:** The data presented in this study are available within the article.

**Conflicts of Interest:** The authors declare no conflict of interest.

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
