# Peer review of "Effect of Hyacinth Treatment by Hydrogen Peroxide Stabilized with Silver and Some Fungicides on the Fungal Infection of Substrate and Bulbs and on Plant Growth and Development"

_agronomy, doi:10.3390/agronomy12112894_

Round 1

Reviewer 1 Report

Dear Authors,

The manuscript "Effect of Hyacinth Treatment on Fungal Development on the Substrate and Bulbs and on Plant Growth and Development" describes the effect of treatment on fungal development on hyacinth bulbs. The material is new and well presented, but a few points require clarification, namely:

Line 139 - “1.1. Treatment and growth stimulation ”- probably you shouldn’t do such a sub-item, it’s confusing. Just continue the text from the paragraph.

The "Introduction" part is quite large. I recommend the authors to revise this part of the text and partially transfer the material to the “discussion”, where they compare the obtained data with previously published experiments in this area.

195 - "Blue Pearl hyacinth" replace the name of the plant with the correct taxon designation, namely:

Hyacinthus orientalis hybrid 'Blue Pearl' or Hyacinthus orientalis 'Blue Pearl'. Indicate where the planting material was harvested, who identified it. Also indicate the size of the bulbs.

276-282 - Indicate the reference, where the methods for calculating the signs of growth and quality of plants are indicated.

Pay attention to the list of references, it needs to be corrected.

Round 2

Reviewer 2 Report

I have reviewed the revised manuscript and still feel that my concerns have not been adequately addressed. 
